# Differences in Ct Values in qPCR Tests for the Diagnosis of Mpox: Results of a Cross-Sectional Study

**DOI:** 10.3390/microorganisms13061355

**Published:** 2025-06-11

**Authors:** Clara Esperanza Santacruz Tinoco, Porfirio Felipe Hernández Bautista, David Alejandro Cabrera Gaytán, Julio Elias Alvarado Yaah, Bernardo Martínez Miguel, Yu-Mei Anguiano Hernández, Alfonso Vallejos Parás, José Esteban Muñoz Medina, Lumumba Arriaga Nieto, Leticia Jaimes Betancourt, Yadira Pérez Andrade

**Affiliations:** 1Coordinación de Calidad de Insumos y Laboratorios Especializados, Instituto Mexicano del Seguro Social, Mexico City 07760, Mexico; clara.santacruz@imss.gob.mx (C.E.S.T.); porfirio.hernandez@imss.gob.mx (P.F.H.B.); julio.alvaradoy@imss.gob.mx (J.E.A.Y.); bernardo.martinezm@imss.gob.mx (B.M.M.); yu.anguiano@imss.gob.mx (Y.-M.A.H.); jose.munozm@imss.gob.mx (J.E.M.M.); 2Coordinación de Vigilancia Epidemiológica, Instituto Mexicano del Seguro Social, Mexico City 03100, Mexico; alfonso.vallejos@imss.gob.mx (A.V.P.); lumumba.arriaga@imss.gob.mx (L.A.N.); yadira.perezan@imss.gob.mx (Y.P.A.); 3Unidad de Medicina Familiar No. 7, Instituto Mexicano del Seguro Social, Mexico City 01400, Mexico; leticia.jaimesb@imss.gob.mx

**Keywords:** mpox, qPCR, epidemiological surveillance, Ct

## Abstract

There is little evidence regarding whether Ct values are influenced by age, type of care, and the timing from the date of symptom onset to obtaining samples. The aim of this study was to determine the differences in the qPCR Ct values for mpox by age, type of care, and sampling opportunity. This descriptive, retrospective study analyzed the qPCR Ct values for mpox. The odds of a low Ct value were assessed by age group, type of care, sampling opportunity, and type of exudate. To determine the odds of a low Ct value, a multivariate bivariate analysis of the variables was performed, and a logistic regression model was developed. A total of 520 positive mpox tests were identified; the general median Ct value was 23.67. There was a difference in Ct values between inpatients and outpatients. Pharyngeal exudate samples had the highest Ct value (34.24), and pustule and scab exudate samples had the lowest values. Samples with a low Ct value (OR = 2.45, 95% CI: 1.13–5.29) were associated with hospital care. Low Ct values were more likely for samples from individuals being cared for in a hospital setting and with pustule skin lesions.

## 1. Introduction

Human monkeypox (mpox) is a zoonotic disease caused by the monkeypox virus, which has been endemic in the rainforest regions of Central and West Africa [1]. The first reported outbreak of the disease outside of Africa [2] was related to the importation of infected mammals in 2003 to the United States of America (USA). Since 2018, as of the end of 2021, 12 cases of mpox have been reported in Europe associated with travel outside of Africa [3].

The incubation period for mpox ranges from 5 to 21 days [4]. The clinical symptoms consist mainly of different skin lesions, accompanied by fever, myalgia, and lymphadenopathy [5]. Patients are likely to be contagious from the onset of symptoms until all dermal lesions have resolved, and during this period, transmission of the mpox virus can occur through direct and close contact through droplets, bodily fluids, or fomites [5].

Mpox was defined as a public health emergency of international concern (PHEIC) by the World Health Organization (WHO) on 23 July 2022 due to the increase and expansion of cases worldwide [6].

Regarding the global context, since 1 January 2022, cases of mpox have been reported in 111 countries, territories, and areas in the six WHO regions; as of 2 May 2023, 87,301 confirmed cases and 130 deaths have been reported worldwide. The peak of the epidemic occurred between week 28 and 32 of 2022. Initially, the cases were detected in Europe (United Kingdom), expanding through the Mediterranean countries and from there to the American continent, where the USA, Brazil, Colombia, and Mexico had the highest number of cases; eventually, the number of cases in the Americas region surpassed that in Europe [3]. Currently, there is a downward trend in cases throughout the world, including in Mexico, where the first case of mpox was identified in Mexico City, with clinical manifestations initiating on 20 May 2022 and laboratory confirmation on 27 May [7].

qPCR is considered the gold standard for the diagnosis of mpox [8] and is used in the epidemiological surveillance system in Mexico [9]. The number of cycles needed for the fluorescent signal to be detected is the cycle threshold (Ct). To assess the presence of viruses in a sample, a qPCR Ct value needs to be established for a sample to be considered positive. Ct is a semiquantitative value inversely related to the amount of RNA in the sample; therefore, a low Ct value indicates a higher viral load [10]. In a study to validate the use of qPCR for different viral cultures, by assessing viral quantities in clinical samples, cycle threshold (Ct) values ranged from 14.13 to 31.80 for dry swab samples and from 11.34 to 34.01 for VTM samples. Therefore, regarding the use of Ct values as markers of viral load, a Ct value > 30 is associated with the limit for confirming in vitro infectivity [11].

Although qPCR is the gold standard for mpox detection, some notable testing conditions and results have been documented; for example, the greater the number of dilutions, the higher the Ct value in different types of samples, with the lowest Ct value (17.59) for an undiluted skin sample [12]; age affects results [13] and the type of sample influences the Ct value [14]. However, there is little evidence regarding whether Ct values are influenced by care setting and age stratification and the timing from the date of symptom onset to obtaining samples. Similarly, studies have focused on the Ct value [10,11,12,13,14] but not on the analysis of Ct RP and the determination of delta Ct by the 2−∆∆CT method. The hypothesis was that there are differences in the qPCR Ct values for mpox by demographic factors and type of care. On that basis, the objective of this study was to determine the differences in qPCR Ct values for mpox using samples from the laboratory-based epidemiological surveillance system, which operates in the Mexican Institute of Social Security (Instituto Mexicano del Seguro Social—IMSS, by its acronym in Spanish), from November 2022 to May 2023. The Laboratory Epidemiological Control System (Sistema de Control Epidemiológico para Laboratorio—SISCEP, by its acronym in Spanish) operates in all the medical units of the IMSS, where the operational user is mainly a person who specializes in epidemiology and retrieves sample data; the laboratory staff produces reports using demographic data and internal tools.

## 2. Materials and Methods

### 2.1. Sample Collection

This was a retrolective, retrospective, multicenter, national-based study. The records were obtained from the SISCEP, which operates in all IMSS medical units. The records of mpox laboratory samples between November 2022 and May 2023 were selected; the following data were recorded: age, sex, state, type of care, nominal laboratory result, and Ct value reported in the epidemiological surveillance system. The samples from the patients (exudate from dermal, pharyngeal, or nasopharyngeal lesions) met the operational definitions of sectoral cases. (A) Probable case: person of any age and sex, presenting one or multiple skin (macula, papule, gallbladder, pustule, and/or scab) or mucosa lesions and who does not have a clinical diagnosis that explains the current picture and one or more of the following signs or symptoms: fever, myalgia, headache, lymphadenopathy, asthenia, arthralgia, and low back pain. In immunocompromised people, the presence of one or multiple skin (macula, papule, gallbladder, pustule and/or scab) or mucosa lesions without the presence of other signs or symptoms can be considered a probable case. (B) Laboratory-confirmed case: a probable case with a positive result for simian pox virus processed by the InDRE or by a verified laboratory with personnel trained to provide a diagnosis. (C) Contact: a person who has had one or more of the following exposures with a probable or confirmed case in the last 21 days: direct skin-to-skin physical contact, including sexual contact, inhalation of respiratory droplets from infected persons, contact with material from skin lesions or mucosa (e.g., scabs), and contact with fomites or contaminated materials, such as clothing, bedding, and utensils for personal use without the appropriate personal protective equipment (PPE) [7].

The sample analyzed was the sample used for the diagnosis of mpox, as an individual could have had more than one sample. Samples that were positive for mpox by qPCR were selected regardless of the Ct value. All samples from patients were captured by the laboratory epidemiological surveillance system for mpox.

### 2.2. Diagnostic Techniques

Samples of pharyngeal, vesicle, pustule, and scab exudates from patients with suspected mpox were collected in the IMSS medical units and sent for analysis under triple packaging as category A samples (UN2814). The samples were filtered after being received as adequate or rejected according to the clinical–epidemiological variables and quality criteria, such as patient identification data, type of sample, and integrity of the primary container. Samples were lysed and nucleic acids were extracted in a class II A2 biosafety cabinet according to the manufacturer’s (QIAGEN, Hilden, Germany) instructions for the QIAamp Blood Mini Kit, cat. 51106. For qPCR, TaqMan Gene Expression Master Mix, cat. 4370074, was used, adding the primers and probe for the target fragment (Table 1). A QuantStudio 5 real-time qPCR system configured with a step program was used to amplify nucleic acids by qPCR (Appendix A).

For interpretation, for all markers, true amplification was considered a Ct value lower than 43, with a sigmoidal profile and delta of fluorescence greater than 200,000. The analytical framework was consistent with the guidelines of the health authority in Mexico [15], and the transferred method is an adaptation of the method described by Li et al. [16].

Also, the 2−∆∆CT method was calculated, and this is a relative quantification strategy for the results of a qPCR or RT-qPCR, which uses the generated Ct, assuming an amplification efficiency of 100% in the analyzed samples. The two “deltas” present in the name of this method refer to the fact that the expression level of a target sample is compared to a control or reference sample, also using a reference gene as a normalizer. The results of this method are usually reported in increments from one sample to the other; however, in this work, we only subtract the Ct of the endogenous gene (RP) from the Ct of the amplification of mpox. Subsequently, using the resulting Ct of each sample (∆CT), we analyzed the means for comparisons between groups (sex, sample site, and attention).

### 2.3. Statistical Analysis

For the univariate analysis, simple frequencies and measures of central tendency and dispersion were calculated. Type of care was divided into outpatient and inpatient. The exudate samples were analyzed by origin, such as pharyngeal and dermal lesions (gallbladder, pustule, and scab). Ages were divided at ≤29 years, 30 to 49 years, and >49 years. The samples were divided into days from the date of onset of symptoms and the sampling in ≤4 days, 4 to 7 days, and >7 days. The marker MPXV was used as a reference to determine the difference in Ct values from the median, and the odds ratio for the variables was calculated. Ct values were divided into <20, 20 a 22, 23 a 25, and 25 and more. In addition, univariate and correlation analyses of time and age and Kruskal–Wallis and Hartley tests were performed. The odds of obtaining a low Ct value were determined by age group, type of care, sampling opportunity, and type of exudate. The median Ct of each variable was taken as the cutoff point. Multivariate analysis and a logistic regression model were used to determine the odds of a low Ct value.

## 3. Results

A total of 520 records of positive mpox tests were identified in the study period. Males represented 93.3% of the samples (485 samples) and females represented 6.7% (35 samples). Among the samples of women, none were recorded as being pregnant at the time of sample collection. The average age of the individuals with positive tests was 33.1 years old (min 14, max 84 years). The median age among women was 31 (SD = 12.4), while among men, it was 32 (SD = 8.5); (−1; CI95% −5.32233–3.32233, *p* < 0.005). Three samples were from children under 18 years; by age group, they were distributed as follows: 202 (38.7%) were less than or equal to 29 years of age, 293 (56.1%) were between 30 and 49 years, and 27 samples (5.2%) were over 49 years of age. By deployed age, 27, 28, and 33 years were the most frequent (18.7%). The samples were from 26 (out of 32) states in the country, mostly from Mexico City, Quintana Roo, the State of Mexico, and Yucatán (59.9%). Most of the samples analyzed came from patients whose management was on an outpatient basis (492), and 28 cases required hospitalization.

The median time from symptom onset to taking the exudate sample was 6.0 days; by sample: oropharynx was 5.5 days; vesicles and pustules was 6 days; and scabs was 7.0 days. The general median mpox virus Ct was 23.67, with a minimum of 13.75 and a maximum of 42.85. There was a difference in Ct values between hospitalized patients and outpatients (21.56 vs. 23.73, *p* = 0.036).

The age group with the lowest Ct value was 30–49 years (Ct value = 23.3; *p* = 0.0248), which was lower than that of the other groups. However, the age group with the highest Ct value was >49 years. A lower Ct was obtained for samples obtained between 4 and 7 days from the onset of symptoms (Ct = 23.15; *p* = 0.0273). Pharyngeal exudate samples had the highest Ct value (34.24), and pustule (23.23) and crust (23.73) exudates had the lowest values (Table 2).

Two Ct groups were formed based on the median, and the odds ratio was calculated for each variable (Table 3). The lowest Ct values were obtained for samples from individuals receiving in-hospital care (OR = 2.45, 95% CI: 1.13–5.29). Younger age increased the probability of being hospitalized; however, in all age groups, age was a protective factor for the value estimate (Table 4).

Finally, multivariate analysis was performed with the median Ct cutoff point; there was an increased risk of hospitalization with a Ct value lower than 22 (OR = 2.31, 95% CI, 1.05–5.07). The odds of a Ct value less than 22 for pharyngeal exudate samples were low (Table 5). The West African clade was present in all samples.

The 2−∆∆CT method indicated increments from hospitalized patients (Figure 1), males (Figure 2), and pustules (Figure 3). For further details, see the Appendix A.

## 4. Discussion

A retrospective analysis was performed to determine the differences in qPCR Ct values for the diagnosis of mpox in the population with social security coverage in Mexico. There was an association between Ct and the type of care, sample type, sampling opportunity, and patient age. Regarding the type of care, the proposed hypothesis was that there is a greater risk of being hospitalized with a lower Ct value; the evidence obtained in this study supported the hypothesis.

The median age reported in mpox samples in Spain was 35 years, while in the present study, it was 33.1; although they are different studies and populations, they both showed that men were predominant and that the predominant age was in the third decade of life [14]. The age group with the lowest Ct was 30–49 years; in 2023, Martins et al. [13] reported a direct correlation between viral load and age. A lower Ct value was also observed for skin samples, especially pustule samples, compared with that for pharyngeal exudate samples, a finding that was previously documented [14,17], which reaffirms the high risk of infection through contact with skin lesions. Ct values were lower for samples obtained 4 to 7 days after symptom onset, a finding that is consistent with the cytopathic effects of the virus detected at five days [17].

In our study, the general median mpox virus Ct was 23.67 versus 30.7 for Callaby et al.; however, the number of samples in that study was smaller (169) and included samples from asymptomatic people, hence the higher Ct value [18]. In a study with a larger number of samples (4233), the average Ct found in positive samples was 20.5, a figure similar to our results [19].

A study carried out in Brazil identified that the opportunity for obtaining samples occurred earlier for children than for adults [13]; therefore, their findings show a moderate-to-high impact on the increase in viral load with age. People under 18 years of age with mpox have higher Ct values than adults, which could indicate lower infectivity and a reduced ability to transmit the disease. Likewise, in a previous study, the viral loads in skin lesions were significantly higher than those in pharyngeal or nasopharyngeal exudate samples (median Ct 22.0 versus 29.0, *p* = 0.0013 and median Ct 22.0 versus 36.5, *p* = 0.0001, respectively) [17], findings that are consistent with the results of a meta-analysis. In dermal lesions, the estimated positivity was 99.9% (95% CI 99.1–100.0; I^2^ = 0%), and that of nasopharyngeal lesions was 86.3% (95% CI 72.4–96.5; I^2^ = 81.6%) [14]; the results were similar for Ct values, with lower values in skin; therefore, samples with a Ct value ≥ 33 represent low-infectious or noninfectious disease [17]. In November 2023, the Democratic Republic of the Congo reported to the WHO the highest number of annual cases ever reported, with new cases in geographic areas that had previously not reported mpox [20]. Subclade Ib was recently identified, and little by little, it expanded to other African countries (Burundi, Democratic Republic of the Congo, Kenya, Rwanda, Uganda, and Zambia) with sustained human-to-human transmission of the virus [21]. Recently, clade Ib mpox was identified in California in a traveler from the USA who returned from East Africa [22]. In Mexico, laboratory-confirmed cases correspond to clade II [23].

Although there are other types of clinical samples for diagnosis (saliva, semen, feces, and urine), they have been shown to yield inaccurate results or to be of low utility [14]; therefore, Mexico has opted for exudate samples from dermal lesions. Importantly, in the present study, the lowest Ct values were measured in patients who needed hospital care in the period 4 to 7 days after the onset of clinical manifestations and in dermal lesions rather than in pharyngeal lesions. In our study, the added value compared to other studies is the use of the ΔΔCt method to eliminate sampling bias by normalizing Ct values for mpox with the endogenous RP gene, revealing that the most clinically useful sample for diagnosis was pustules, with three times more viral genetic material, followed by biopsies; however, this sample is not representative due to the quantity included in this study. And given the quality of care, hospitalized patients were found to have 1.80 times more viral material than outpatients. Importantly, our findings showed a direct association between the Ct value measured in the first week after onset, as Mazzotta has documented [24].

Although the Ct values for mpox have been documented by days of collection and by type of sample [14,17], the contribution of this study is the breakdown by type of care and age with a determination of the odds of a low Ct value, which has been omitted or studied in an ephemeral way in other studies, with a greater number of records involved. Therefore, the strengths of the study are as follows: (1) The Ministry of Health, through The Institute of Epidemiological Diagnosis and Reference (InDRE), has designated the State Laboratories of Public Health (Laboratorios Estatales de Salud Pública—LESP, by its acronym in Spanish) and the Central Laboratory of Epidemiology of the IMSS as facilities qualified for diagnosing mpox and thus the endorsement of the authorities [7,15]. (2) Samples of exudates of different types from various medical units in different states of the country were analyzed.

Regarding state representation, the majority of the samples came from the center of the country and the Yucatan Peninsula, a finding that is consistent with the national incidence rates of mpox [7]. At the same time, it is emphasized that the Yucatan Peninsula (States of Campeche, Quintana Roo, and Yucatan) has a large influx of tourists and commercial activities in the areas surrounding Merida and Cancun, involving population movement for recreational purposes in the summer or the behavior of the Southern Hemisphere.

This study explored the factors associated with mpox infection and their association with the viral load in bodily fluids and the fold increase situation that has not been documented in previous studies (2−∆∆CT method). However, this study has limitations due to the sample size. However, the results were statistically significant. In the multivariate analysis, only hospitalization and pharyngeal exudate were statistically significant with regard to increased odds of a low Ct value. Another limitation was the potential differential bias in a larger number of outpatient samples; the type of lesion was described but the anatomical site was ignored, as it has been previously reported that there is no difference in the Ct value and culture results among anatomical sites [17]. The clinical conditions of the patients are unknown; one of the most important is the state of immunocompromise and whether a patient is infected with HIV in order to compare against the results of other studies [14,17]. Finally, other studies have taken various samples over time from patients with mpox; in the present study, the samples were taken at a single moment (at the time of identification as a new case) [13,14].

## 5. Conclusions

Low Ct values were obtained for samples from individuals receiving in-hospital care for samples from pustule and scab skin lesions and from individuals in the 30–49-year age group. The best sample was pustules by the 2−∆∆CT method.

## Figures and Tables

**Figure 1 microorganisms-13-01355-f001:**
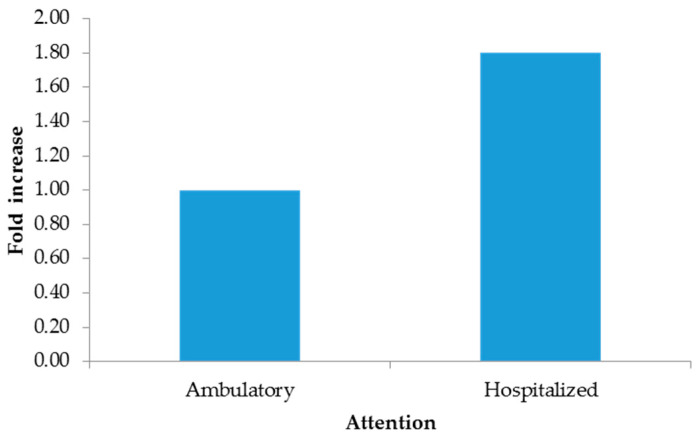
Fold increase by attention.

**Figure 2 microorganisms-13-01355-f002:**
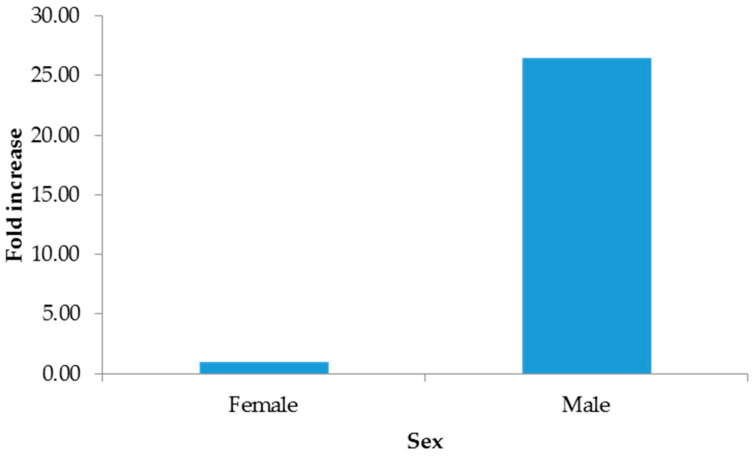
Fold increase by sex.

**Figure 3 microorganisms-13-01355-f003:**
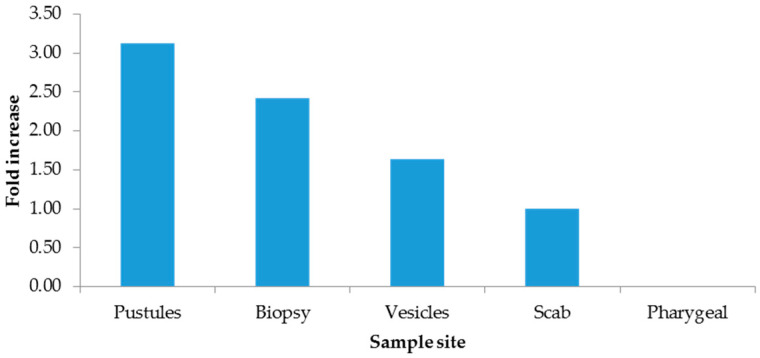
Fold increase by sample site.

**Table 1 microorganisms-13-01355-t001:** Indicators, probes, and sequences.

Target Fragment	Initiator	Sequence
	Probe	
G2_R assay (generic MPXV detection)	MPXV-F	5′-GGAAAATGTAAAGACAACGAATACAG-3′
	MPXV-R	5′-GCTATCACATAATCTGGAAGCGTA-3′
	MPXV-P	5′-FAM-AAGCCGTAATCTATGTTGTCTATCGTGTCC-3′BHQ1
G2_WA assay (detection of West African clade strains)	WA-F	5′-CACACCGTCTCTTCCACAGA-3′
	WA-R	5′-GATACAGGTTAATTTCCACATCG-3′
	WA-P	5′-FAM-AACCCGTCGTAACCAGCAATACATTT-3′BHQ1
G2_CC assay (detection of the Congo Basin clade strains)	CC-F	5′-TGTCTACCTGGATACAGAAAGCAA-3′
	CC-R	5′-GGCATCTCCGTTTAATACATTGAT-3′
	CC-P	5′-FAM-CCCATATATGCTAAATGTACCGGTACCGGA-3′BHQ1
Ribonuclease P of human origin (internal control)	RP-F	5′-AGATTTGGACCTGCGAGCG-3′
	RP-F	5′-GAGCGGCTGTCTCCACAAGT-3′
	RP-P	5′FAM-TTCTGACCTGAAGGCTCTGCGCG-3′BHQ1

**Table 2 microorganisms-13-01355-t002:** Description of the Ct values according to the clinical–epidemiological characteristics of the mpox samples.

Variables	Med Ct	Min	Max	KW ^1^	*p*
All	23.67	13.75	42.85		
Type of care					
Ambulatory	23.73	13.75	42.85		
Hospitalized	21.56	16.53	40.52	4.3987	0.036
Exudate samples					
Scab	23.73	15.9	41.89	0.009	0.92
Pharyngeal	34.24	19.11	42.85	21.55	<0.001
Pustule	23.23	13.75	40.52	7.94	0.004
Vesicle	24.01	16.16	42.21	2.17	0.14
Sampling					
≤4 days	24.00	14.61	41.89	0.9564	0.3281
4 to 7 days	23.15	13.75	40.96	4.8706	0.0273
>7 days	24.08	16.53	42.85	2.4181	0.1199
Age					
≤29 years	23.93	16.16	42.85	2.7881	0.95
30 to 49 years	23.30	13.75	42.21	5.0392	0.0248
>49 years	25.24	17.96	39.55	1.8951	0.1686

^1^ Kruskal–Wallis for two groups.

**Table 3 microorganisms-13-01355-t003:** Odds of having a low Ct value for mpox samples.

Variable	OR	95%CI
Hospitalized	2.45	1.13–5.29
Pustule	1.57	1.10–2.23
Pharyngeal	0.11	0.04–0.32
4 to 7 days of sampling ^1^	1.46	1.03–2.07
30 to 49 years ^1^	1.58	1.09–2.28

^1^ The median Ct of the variable was taken as the cutoff point.

**Table 4 microorganisms-13-01355-t004:** Trend of Ct values for mpox samples according to type of medical care.

Ct	Hospitalized	Ambulatory	Total	OR	95%CI
<20	11	91	102	1	
20 to 22	4	80	84	0.41	0.12–1.35
23 to 25	6	133	139	0.37	0.13–1.04
25 and more	7	188	195	0.3	0.11–0.82

χ^2^_trend_ = 5.72, *p* = 0.01676.

**Table 5 microorganisms-13-01355-t005:** Multivariate analysis of the odds of having a low Ct value for mpox samples.

Variable	OR	95%CI
Hospitalized	2.31	1.05–5.07
Pharyngeal	0.12	0.01–0.92
4 to 7 days of sampling ^1^	1.19	0.82–1.72
30 to 49 years ^1^	1.59	1.09–2.30

^1^ Ct 22 was taken as the cutoff point for the logistic regression model.

## Data Availability

Data Availability Statements are available in https://doi.org/10.6084/m9.figshare.28696490.v1.

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
