# Peer review of "Differences in Ct Values in qPCR Tests for the Diagnosis of Mpox: Results of a Cross-Sectional Study"

_microorganisms, 2025, doi:10.3390/microorganisms13061355_

Round 1
Reviewer 1 Report
Comments and Suggestions for Authors
This study investigated that the qPCR Ct values for mpox as an indicator of viral load, infectivity, and the risk of transmission depend on age of patient, type of care and type of sample.
However, the authors should more clearly suggest the possible directions for further applications.
Author Response
REVIWER 1
Comments and Suggestions for Authors
This study investigated that the qPCR Ct values for mpox as an indicator of viral load, infectivity, and the risk of transmission depend on age of patient, type of care and type of sample.
However, the authors should more clearly suggest the possible directions for further applications.
A = The observation is addressed in the methodology, results, and discussion sections. A methodology was added that has greater clinical implications in the samples by determining the 2−∆∆CT. (Page 3).
Reviewer 2 Report
Comments and Suggestions for Authors
Clara Esperanza Santacruz Tinoco etal,to determine the differences in the qPCR Ct values for mpox by age, type of care and sampling opportunity. The result shows that Low Ct values were more likely for samples from individuals being cared for in a hospital setting and with scab and pustule skin lesions. The statistical results are of great significance for the treatment of monkeypox virus. Several issues that need to be attention.
- Has the difference in the pathogenicity of the monkeypox virus in different regions been taken into account
- The ct value test for monkeypox is usually conducted at a hospital after a period of viral infection or when symptoms appear, as the proliferation ability of the monkeypox virus declines.
- Viral nucleic acid technology has indeed achieved great success. Is it too simplistic to judge the infection situation solely based on ct values?
- The author cited relatively few literatures and suggested appropriately increasing the discussion on the changes in nucleic acid levels during monkeypox infection
Author Response
REVIWER 2
Comments and Suggestions for Authors
Clara Esperanza Santacruz Tinoco et al,to determine the differences in the qPCR Ct values for mpox by age, type of care and sampling opportunity. The result shows that Low Ct values were more likely for samples from individuals being cared for in a hospital setting and with scab and pustule skin lesions. The statistical results are of great significance for the treatment of monkeypox virus. Several issues that need to be attention.
- Has the difference in the pathogenicity of the monkeypox virus in different regions been taken into account.
A = Determining the pathogenicity of the virus was not the scope of this study. However, the incidence rate of the disease has been observed most frequently in the Yucatán Peninsula (a tourist area) and Mexico City. (Page 8).
- The ct value test for monkeypox is usually conducted at a hospital after a period of viral infection or when symptoms appear, as the proliferation ability of the monkeypox virus declines.
A = In our study, as with the country's epidemiological situation, most cases were managed as outpatients. However, an addition was made to the methodology (2−∆∆CT method) to determine the most significant clinical sample, as well as the type of care, as reflected in the methodology, results, figures, discussion, and supplementary material sections.
- Viral nucleic acid technology has indeed achieved great success. Is it too simplistic to judge the infection situation solely based on Ct values?
A= Your observation is correct; that is why the determination of 2−∆∆CT method. (Page 3).
- The author cited relatively few literatures and suggested appropriately increasing the discussion on the changes in nucleic acid levels during monkeypox infection.
A = Paragraphs are added to the discussion. (Page 7 and 8).
Reviewer 3 Report
Comments and Suggestions for Authors
This study investigates the relationship between Ct values in qPCR for MPXV and several clinical/epidemiological factors (age, sample timing, type of care, and sample type). The topic is relevant, providing meaningful data on diagnostic value interpretation. However, the manuscript would benefit from structural clarifications, refinement in writing, and more rigorous statistical interpretation.
Major concerns:
- The introduction should be improved. The novelty is modest as similar studies have been reported, particularly regarding sample type and timing. The added value could be the association with care setting and age stratification. Please, strengthen the novelty by better specifying how this dataset provides insight not yet found in previous literature.
- Clarify the retrospective/retrolective nature of the present study: explain inclusion/exclusion criteria more explicitly and describe data processing steps in detail.
- Clarify which qPCR assay was used for the main analysis and whether different clade detection was relevant to interpretation.
- The statistical analysis lacks a detailed explanation. For example, the reasoning behind cut-offs (e.g., Ct <22) is not justified.
- Revise tables for clarity: for example, Table 4 is unclear; include total Ns and clarify how categories were derived.
Author Response
REVIWER 3
Comments and Suggestions for Authors
This study investigates the relationship between Ct values in qPCR for MPXV and several clinical/epidemiological factors (age, sample timing, type of care, and sample type). The topic is relevant, providing meaningful data on diagnostic value interpretation. However, the manuscript would benefit from structural clarifications, refinement in writing, and more rigorous statistical interpretation.
Major concerns:
- The introduction should be improved. The novelty is modest as similar studies have been reported, particularly regarding sample type and timing. The added value could be the association with care setting and age stratification. Please, strengthen the novelty by better specifying how this dataset provides insight not yet found in previous literature.
A = It was included in the introduction and discussion. (Page 7 and 8).
- Clarify the retrospective/retrolective nature of the present study: explain inclusion/exclusion criteria more explicitly and describe data processing steps in detail.
A = It was added to the methodology. (Page 3).
- Clarify which qPCR assay was used for the main analysis and whether different clade detection was relevant to interpretation.
A = In all mpox-positive samples, the West African clade was detected. (Page 5).
- The statistical analysis lacks a detailed explanation. For example, the reasoning behind cut-offs (e.g., Ct <22) is not justified.
- A = The Ct value of 22 was the one that had the best statistical significance in the bivariate and multivariate analysis.
- Revise tables for clarity: for example, Table 4 is unclear; include total Ns and clarify how categories were derived.
A = The total is added to the table.
Round 2
Reviewer 2 Report
Comments and Suggestions for Authors
Paper Acceptance
Reviewer 3 Report
Comments and Suggestions for Authors
The article is suitable for publication.